# Structure of Fam20A reveals a pseudokinase featuring a unique disulfide pattern and inverted ATP-binding

Jixin Cui[1†], Qinyu Zhu[2,3†], Hui Zhang[2,3], Michael A Cianfrocco[4], Andres E Leschziner[4], Jack E Dixon[1,4,5]*, Junyu Xiao[3]*

[1]Department of Pharmacology, University of California, San Diego, United States; [2]Academy for Advanced Interdisciplinary Studies, Peking University, Beijing, China; [3]The State Key Laboratory of Protein and Plant Gene Research, Peking-Tsinghua Center for Life Sciences, Peking University, Beijing, China; [4]Department of Cellular and Molecular Medicine, University of California, San Diego, United States; [5]Department of Chemistry and Biochemistry, University of California, San Diego, United States

*For correspondence: jedixon@ucsd.edu (JED); junyuxiao@pku.edu.cn (JX)

†These authors contributed equally to this work

Competing interests: The authors declare that no competing interests exist.

**Abstract** Mutations in *FAM20A* cause tooth enamel defects known as Amelogenesis Imperfecta (AI) and renal calcification. We previously showed that Fam20A is a secretory pathway pseudokinase and allosterically activates the physiological casein kinase Fam20C to phosphorylate secreted proteins important for biomineralization (Cui et al., 2015). Here we report the nucleotide-free and ATP-bound structures of Fam20A. Fam20A exhibits a distinct disulfide bond pattern mediated by a unique insertion region. Loss of this insertion due to abnormal mRNA splicing interferes with the structure and function of Fam20A, resulting in AI. Fam20A binds ATP in the absence of divalent cations, and strikingly, ATP is bound in an inverted orientation compared to other kinases. Fam20A forms a dimer in the crystal, and residues in the dimer interface are critical for Fam20C activation. Together, these results provide structural insights into the function of Fam20A and shed light on the mechanism by which Fam20A mutations cause disease.

## Introduction

The human kinome contains more than 500 members, which function primarily in the cytoplasm or nucleus to regulate many cellular processes (*Manning et al., 2002*). Approximately 10% of the kinome are considered pseudokinases due to their lack of certain residues typically required for catalysis. Some of these proteins, however, achieve activity using residues located at unconventional positions and are thus *bona fide* kinases (*Min et al., 2004*; *Eswaran et al., 2009*; *Villa et al., 2009*; *Zhu et al., 2016*). Others are completely devoid of catalytic activity but play important scaffolding roles and can serve as allosteric regulators of active kinases (*Boudeau et al., 2006*; *Zeqiraj and van Aalten, 2010*, *Shaw et al., 2014*; *Kung and Jura, 2016*).

Novel kinases specifically residing in the secretory pathway, exemplified by the family of sequence similarity 20 (Fam20) family, have been recently identified to phosphorylate secreted and lumenal proteins and glycans (*Tagliabracci et al., 2013a*, *2013b*; *Sreelatha et al., 2015*). As commonly seen with secretory proteins, these kinases are glycosylated and contain disulfide bonds for structural integrity. Fam20C, the best-characterized member of the Fam20 family, has been established as the physiological casein kinase that phosphorylates the vast majority of the secreted phosphoproteome (*Tagliabracci et al., 2012*, *2015*). Fam20C substrates are involved in a wide spectrum

of biological processes, including the formation of bones and teeth (*Tagliabracci et al., 2012*; *Cui et al., 2015*). Newborns with loss-of-function *FAM20C* mutations present with deadly sclerosing osteomalacia with cerebral calcification known as Raine syndrome (*Simpson et al., 2007*; *Whyte et al., 2017*). Patients with hypomorphic *FAM20C* mutations can survive but manifest various anomalies, including tooth enamel formation defect referred to as Amelogenesis Imperfecta (AI), hypophosphatemia, and ectopic calcification (*Fradin et al., 2011*; *Rafaelsen et al., 2013*; *Takeyari et al., 2014*; *Acevedo et al., 2015*; *Elalaoui et al., 2016*). The crystal structure of the Fam20C ortholog from *Caenorhabditis elegans* (ceFam20) revealed an atypical kinase architecture and key catalytic residues that are uniquely present in the Fam20 family of kinases (*Xiao et al., 2013*).

Fam20B and Fam20A are two paralogs of Fam20C in vertebrates (*Nalbant et al., 2005*). Fam20B phosphorylates a xylose residue during the biosynthesis of proteoglycans and plays a key role in regulating glycan elongation (*Koike et al., 2009*; *Wen et al., 2014*). Mutations of Fam20A result in AI and ectopic calcification such as nephrocalcinosis, analogous to the defects caused by non-lethal Fam20C mutations (*Volodarsky et al., 2015*; *Cherkaoui Jaouad et al., 2015*, *Wang et al., 2014*, *2013*; *O'Sullivan et al., 2011*; *Kantaputra et al., 2014b*, *2014a*; *Jaureguiberry et al., 2012*; *Cho et al., 2012*; *Cabral et al., 2013*; *Kantaputra et al., 2017*). We have previously demonstrated that Fam20A lacks an essential residue for catalysis and is therefore the first pseudokinase identified in the secretory pathway (*Cui et al., 2015*). Fam20A forms a functional complex with Fam20C and allosterically increases Fam20C activity towards secretory substrates, including the enamel matrix proteins, whose phosphorylation is critical for enamel formation (*Cui et al., 2015*). As compared to other Fam20 family members, Fam20A has a unique and highly conserved insertion in the Gly-rich loop (*Xiao et al., 2013*). Truncation of this insertion due to aberrant RNA splicing causes AI, strongly suggesting its role in maintaining Fam20A function (*Cho et al., 2012*). Furthermore, Fam20A binds ATP despite being catalytically inactive (*Cui et al., 2015*), the structural basis and functional impact of which remain elusive.

Here, we report the nucleotide-free and ATP-bound crystal structures of Fam20A. Although the kinase core of Fam20A is structurally similar to that of ceFam20, Fam20A displays an unusual disulfide pattern dictated by a pair of cysteine residues within the unique insertion region. Strikingly, ATP binds to Fam20A in an unprecedented orientation independent of cations. These results reinforce the conclusion that Fam20A is a pseudokinase in the secretory pathway and facilitate a deeper understanding of AI caused by Fam20A mutations.

## Results

### Fam20A displays a unique disulfide pattern

We determined the crystal structure of human Fam20A at 2.5 Å resolution (*Figure 1A*, *Table 1*). The kinase core of Fam20A (residues 160–525) is structurally similar to that of ceFam20, and can be superimposed onto ceFam20 with a rmsd (root-mean-square difference) of 2.1 Å over 322 aligned Cα atoms. Most secondary structures in the kinase core of Fam20A aligned well with the equivalent structural elements in ceFam20 except for three regions: the Kβ1-Kβ2 loop, the Kβ6-Kβ7 loop, and the Kβ3-Kα3 loop (hereafter we use 'K' to denote the secondary structures in the kinase core of the Fam20 proteins, and 'N' to denote the secondary structures in their unique N-terminal segments. *Figure 1A and B*; *Figure 1—figure supplement 1*). An insertion of seventeen residues specifically exists in the Kβ1-Kβ2 loop of Fam20A (*Figure 1—figure supplement 1*, *Figure 1—figure supplement 2*). This insertion forms two short α-helices (Kα2A, Kα2B) and forces an upswing of the Kβ1-Kβ2 loop. The Kβ1-Kβ2 loop in turn levers up the Kβ6-Kβ7 loop and induces the formation of a short helix (Kα5A) in this region. The Kβ3-Kα3 loop also swings up to engage with the Kβ6-Kβ7 loop. Gln258$^{Fam20A}$, which replaces a Glu essential for the catalytic activity of Fam20C (Glu213$^{ceFam20}$), resides at the C-terminal end of the Kβ3-Kα3 loop. As a result of the upswing of this loop, Gln258$^{Fam20A}$ is dislodged from the 'active site' of Fam20A (*Figure 1A*). The N-terminal segment of Fam20A contains two α-helices and adopts a different topology compared to that of ceFam20. A long helix, Nα2, is held alongside the kinase core and is involved in binding to ATP as described below.

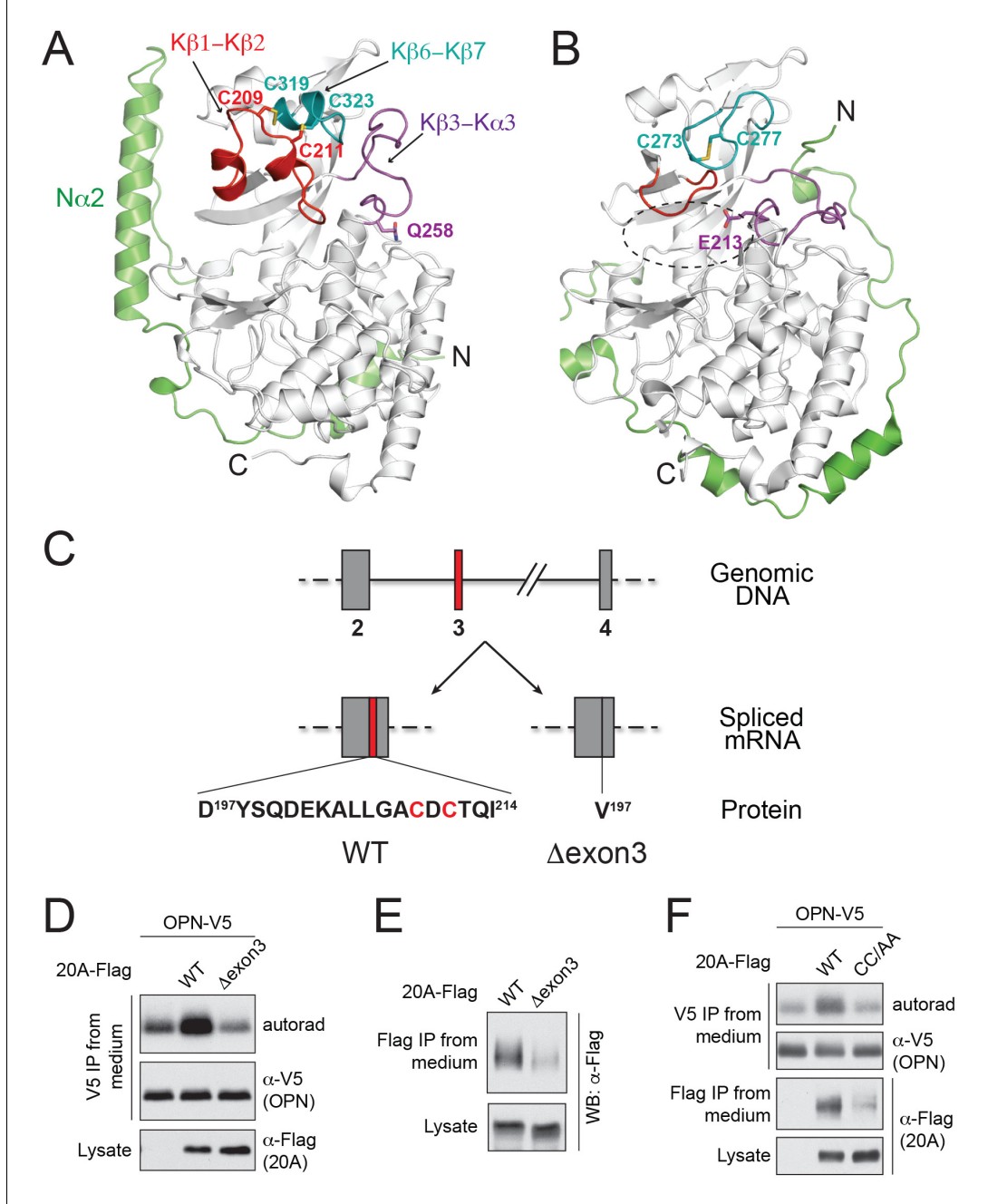

**Figure 1.** The structure of Fam20A reveals a unique disulfide pattern. (**A**) Structure of human Fam20A. The N-terminal segment was colored in green. The Kβ1-Kβ2, Kβ6-Kβ7, and Kβ3-Kα3 loops were colored in red, teal, and magenta, respectively. The rest of the kinase core is shown in white. The side chains of Cys209, Cys211, Cys319, Cys323, and Gln258 are shown as sticks. The N- and C-termini of the molecule are indicated. (**B**) Structure of ceFam20 (PDB ID: 4KQA) is shown in the same orientation as Fam20A in (**A**) and colored in the same color scheme. The side chains Cys273, Cys277, and Glu213 are shown as sticks. The active site of ceFam20 is indicated with an oval in dashed line. (**C**) The schematic of the human Fam20A gene structure and RNA splicing. Exon three is highlighted in red. Relevant amino acid sequences are shown at the bottom. The two Cys residues in the Kβ1-Kβ2 loop are highlighted in red. (**D**) Effect of Fam20A exon three deletion on OPN phosphorylation. C-terminal V5-tagged OPN was expressed alone or co-expressed with Flag-tagged Fam20A WT or Δexon3 in U2OS cells that were metabolically labeled with $^{32}$P orthophosphate. OPN-V5 was immunoprecipitated from the medium. Total OPN and $^{32}$P incorporation were detected by anti-V5 immunoblotting and autoradiography. Expression of Fam20A was monitored by anti-Flag immunoblotting. (**E**) Effect of exon three deletion on Fam20A secretion. Flag-tagged Fam20A WT or Δexon3 was expressed in U2OS cells and immunoprecipitated from conditioned media using anti-Flag antibody. (**F**) Effect of C209A/C211A mutation (CC/AA) on Fam20A secretion and OPN phosphorylation.

*Figure 1 continued on next page*

*Figure 1 continued*

The following figure supplements are available for figure 1:

**Figure supplement 1.** Sequence alignment of Fam20A, Fam20C, and ceFam20 in the kinase core region.

**Figure supplement 2.** Sequence alignment of Fam20A, Fam20B, and Fam20C family members around the Fam20A-specific insertion region.

**Table 1.** Data collection and refinement statistics.

| | Fam20A (PDB ID: 5WRR) | Fam20A with ATP (PDB ID: 5WRS) |
|---|---|---|
| **Data collection** | | |
| Space group | P $3_2$ 2 1 | P $3_2$ 2 1 |
| Cell dimensions | a = 156.854 Å, b = 156.854 Å, c = 143.655 Å | a = 157.202 Å, b = 157.202 Å, c = 144.802 Å |
| Wavelength (Å) | 0.979 | 1.009 |
| Resolution (Å) | 2.50 | 2.75 |
| $R_{merge}$ | 0.172 (1.804) | 0.168 (1.470) |
| $R_{pim}$ | 0.055 (0.566) | 0.053 (0.472) |
| CC1/2 (highest-resolution shell) | 0.841 | 0.788 |
| $I / \sigma I$ | 28.8 (2.8) | 21.2 (2.4) |
| Completeness (%) | 99.9 (100.0) | 99.9 (100.0) |
| Multiplicity | 10.9 (11.1) | 10.7 (10.6) |
| Wilson B-factor | 54.1 | 57.9 |
| **Refinement** | | |
| Reflections used in refinement | 70216 (6906) | 53997 (5329) |
| Reflections used for $R_{free}$ | 1994 (197) | 1946 (193) |
| $R_{work}$ / $R_{free}$ | 0.215/0.231 | 0.203/0.238 |
| No. of non-hydrogen atoms | | |
| Protein | 7039 | 7018 |
| Ligands | 98 | 146 |
| Protein residues | 876 | 874 |
| *B*-factors | | |
| Protein | 75.1 | 72.3 |
| Ligands | 116.7 | 101.4 |
| R.m.s deviations | | |
| Bond lengths (Å) | 0.002 | 0.003 |
| Bond angles (°) | 0.62 | 1.01 |
| Ramachandran | | |
| Favored (%) | 96.8 | 96.5 |
| Allowed (%) | 3.1 | 3.5 |
| Outliers (%) | 0.1 | 0 |

Each dataset was collected from a single crystal. Values in parentheses are for highest-resolution shell.

Fam20A displays a distinct disulfide pattern compared to ceFam20. As we previously described, four pairs of disulfide bonds exist in ceFam20 (*Xiao et al., 2013*). Three of them are also present in Fam20A (*Figure 1—figure supplement 1*). However, a disulfide bond corresponding to Cys273$^{ceFam20}$-Cys277$^{ceFam20}$ in the Kβ6-Kβ7 loop is absent in Fam20A. Instead, Cys209$^{Fam20A}$ and Cys211$^{Fam20A}$ in the Kβ1-Kβ2 loop form two disulfide bonds with Cys319$^{Fam20A}$ and Cys323$^{Fam20A}$ (the two Cys that align with Cys273$^{ceFam20}$ and Cys277$^{ceFam20}$), respectively (*Figure 1A*, *Figure 1—figure supplement 1*). These two disulfide bonds tether the Kβ1-Kβ2 loop to the Kβ6-Kβ7 loop, imposing a strong restraint on the structural flexibility of this region. Notably, Cys209$^{Fam20A}$ and Cys211$^{Fam20A}$ are within the 17-residue insertion in the Kβ1-Kβ2 loop and are specifically conserved in Fam20A orthologs (*Figure 1—figure supplement 2*), suggesting that this unique disulfide pattern of Fam20A is evolutionarily conserved.

The functional importance of these two Fam20A-specific disulfide bonds is further demonstrated by a patient-derived Fam20A splicing variant. The N-terminal portion of the Kβ1-Kβ2 loop, including Cys209$^{Fam20A}$ and Cys211$^{Fam20A}$, is encoded by a small exon, exon 3, of the *FAM20A* gene. In an AI patient with compound heterozygous mutations in *FAM20A* (*Cho et al., 2012*), one allele has an A-G transition at the acceptor site of intron 2 (c.590-2A-G), resulting in exon three skipping. The expressed Fam20A variant has amino acids 197–214 substituted with a Val (Fam20A-Δexon3, *Figure 1C*). Since the other allele has a nonsense mutation (R276X) that abrogates the majority of the kinase domain, Fam20A-Δexon3 is the only form of Fam20A present in the patient that gave rise to the disease. To evaluate the function of Fam20A-Δexon3, we co-expressed V5-tagged osteopontin (OPN) with Flag-tagged Fam20A-WT or Fam20A-Δexon3 in U2OS cells, metabolically labeled the cells with $^{32}$P orthophosphate, and analyzed incorporation of radiolabeled phosphate into the V5-immunoprecipitates. Consistent with our previous observation, ectopic expression of Fam20A-WT significantly enhanced the kinase activity of endogenous Fam20C towards OPN, resulting in a higher level of OPN phosphorylation (*Figure 1D*) (*Cui et al., 2015*). In contrast, expression of Fam20A-Δexon3 failed to increase OPN phosphorylation. Compared with Fam20A-WT, Fam20A-Δexon3 was poorly secreted and thus probably misfolded (*Figure 1E*). Similarly, Fam20A-CC/AA, a Fam20A mutant having both Cys209$^{Fam20A}$ and Cys211$^{Fam20A}$ mutated to Ala, also showed attenuated secretion and diminished ability to enhance OPN phosphorylation (*Figure 1F*). Thus, the unique disulfide pattern in Fam20A plays an important role in maintaining its structure and function.

## Fam20A binds ATP without cations

We have previously demonstrated that both Fam20A and Fam20C could bind ATP by means of thermal stability shift assays (*Cui et al., 2015*). Consistent with the fact that metal ions such as Mn$^{2+}$ are required for the kinase activity of Fam20C, Fam20C bound ATP in the presence of Mn$^{2+}$, whereas ATP alone induced little changes in its melting temperature ($T_m$) (*Figure 2A*). In contrast, ATP alone caused dramatic $T_m$ increase of Fam20A, indicating that ATP binding to Fam20A could be cation-independent (*Figure 2A*).

To determine the binding affinity of Fam20A/ATP interaction, we measured the $ΔT_m$ of Fam20A at different ATP concentrations in the presence or absence of Mn$^{2+}$, and deduced apparent dissociation constant ($K_d^{app}$) values. This method has been validated by Murphy et al. to evaluate the affinity of ATP for a number of kinases (*Murphy et al., 2014b*). Our results show that Fam20A binds ATP with a $K_d^{app}$ of 3 μM in the absence of cations as compared to 156 μM in the presence of saturating amount of Mn$^{2+}$ (*Figure 2B*). This suggests that Fam20A preferably binds ATP without cations.

Fam20A also binds other nucleoside triphosphates including GTP, CTP, and UTP in the absence of metal ions, albeit to lesser degrees compared to ATP (*Figure 2C*). In contrast, Fam20C selectively binds Mn$^{2+}$/ATP. Furthermore, Fam20A prefers binding ATP over ADP (*Figure 2D*), indicating that the γ-phosphate group of ATP is important for the interaction.

The significant $T_m$ increase of Fam20A upon ATP binding suggests that ATP stabilizes Fam20A structure. We wondered whether ATP also stabilizes the Fam20A/Fam20C complex. Recombinantly purified Fam20A and Fam20C were mixed (with Fam20A in slight excess to promote complex formation) and passed through a gel filtration column (*Figure 2E*). In the absence of ATP, the peak for the Fam20A/Fam20C complex appeared asymmetrical with a 'shoulder'. When ATP is present, the peak of the complex became much sharper, suggesting that ATP improved the structural homogeneity of the Fam20A/Fam20C complex.

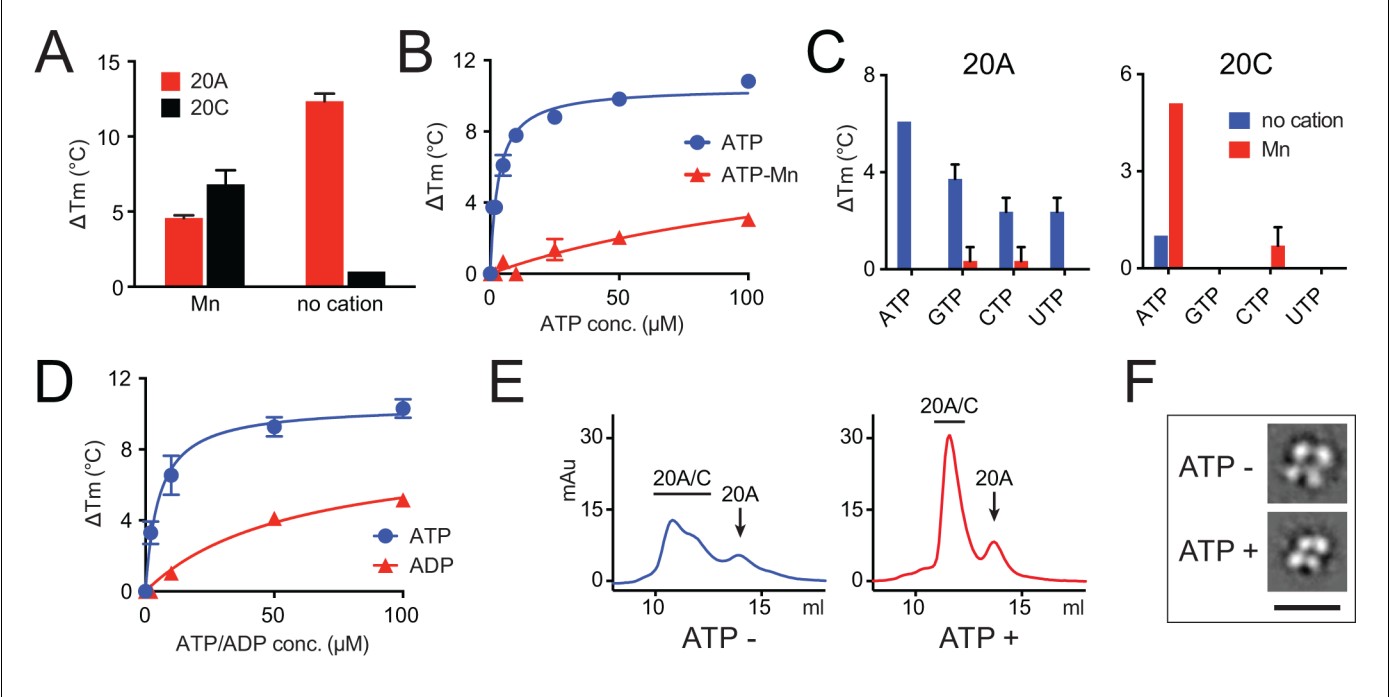

**Figure 2.** Fam20A binds ATP without cations. (**A**) Thermal stability shift assays of ATP binding to Fam20A or Fam20C in the presence or absence of $Mn^{2+}$. ATP, 250 µM; $MnCl_2$, 1 mM. (**B**) Fam20A thermal stability shift assay. The $\Delta T_m$ values are plotted against ATP concentration. Data points are represented as mean ± SD and fitted by non-linear regression of the equation $Y = \Delta T_m max*X/(K_d + X)$. (**C**) Thermal stability shift assays of ATP, GTP, CTP or UTP binding to Fam20A or Fam20C. Nucleotide concentration is 10 µM for Fam20A and 250 µM for Fam20C. (**D**) Thermal stability shift assays of ATP or ADP binding to Fam20A without cations. Data were plotted and fitted as in (**B**). (**E**) Gel filtration analyses of the Fam20A/Fam20C complex in the absence or presence of 250 µM ATP. (**F**) 2D class averages of the negatively stained Fam20A/20C complexes. Representative 2D class averages of the complex assembled in the presence or absence of ATP (250 µM) are shown at the same pixel size to highlight the overall size changes between the two samples. Scale bar, 200 Å.

The following figure supplements are available for figure 2:

**Figure supplement 1.** 2D class averages of the Fam20A/20C complex assembled in the presence or absence of ATP (250 µM) determined by negative-stain single particle electron microscopy.

**Figure supplement 2.** Representative negative stain micrographs of the Fam20A/20C complex assembled in the presence or absence of ATP (250 µM).

To further assess the effect of ATP on the assembly of the Fam20A/Fam20C complex, we performed single particle electron microscopy on negatively stained samples of Fam20A/Fam20C. Two-dimensional (2D) class averages of the complex exhibited a clover-shaped structure that is consistent in size with a Fam20A/Fam20C tetramer (*Figure 2F*). By comparing 2D class averages for each condition, we observed that addition of ATP rendered the Fam20A/Fam20C particles smaller in size (*Figure 2F*, *Figure 2—figure supplement 1* and *Figure 2—figure supplement 2*), indicating a more compact and less flexible conformation of the complex. Collectively, these results suggest that ATP binding to Fam20A stabilizes Fam20A as well as the Fam20A/Fam20C complex.

## ATP binds to Fam20A in an inverted orientation

To elucidate how Fam20A binds ATP, we soaked ATP into the Fam20A crystals obtained above and determined the complex structure at 2.75 Å resolution (*Figure 3A*, *Table 1*). Surprisingly, ATP binds to Fam20A in an unusual orientation in the structure. In an independent experiment, we also co-crystallized Fam20A with ATP and determined the structure at 2.9 Å. The co-crystal structure is almost identical to the soaked one, suggesting that the unexpected ATP-binding mode is not an artifact caused by the soaking experiment.

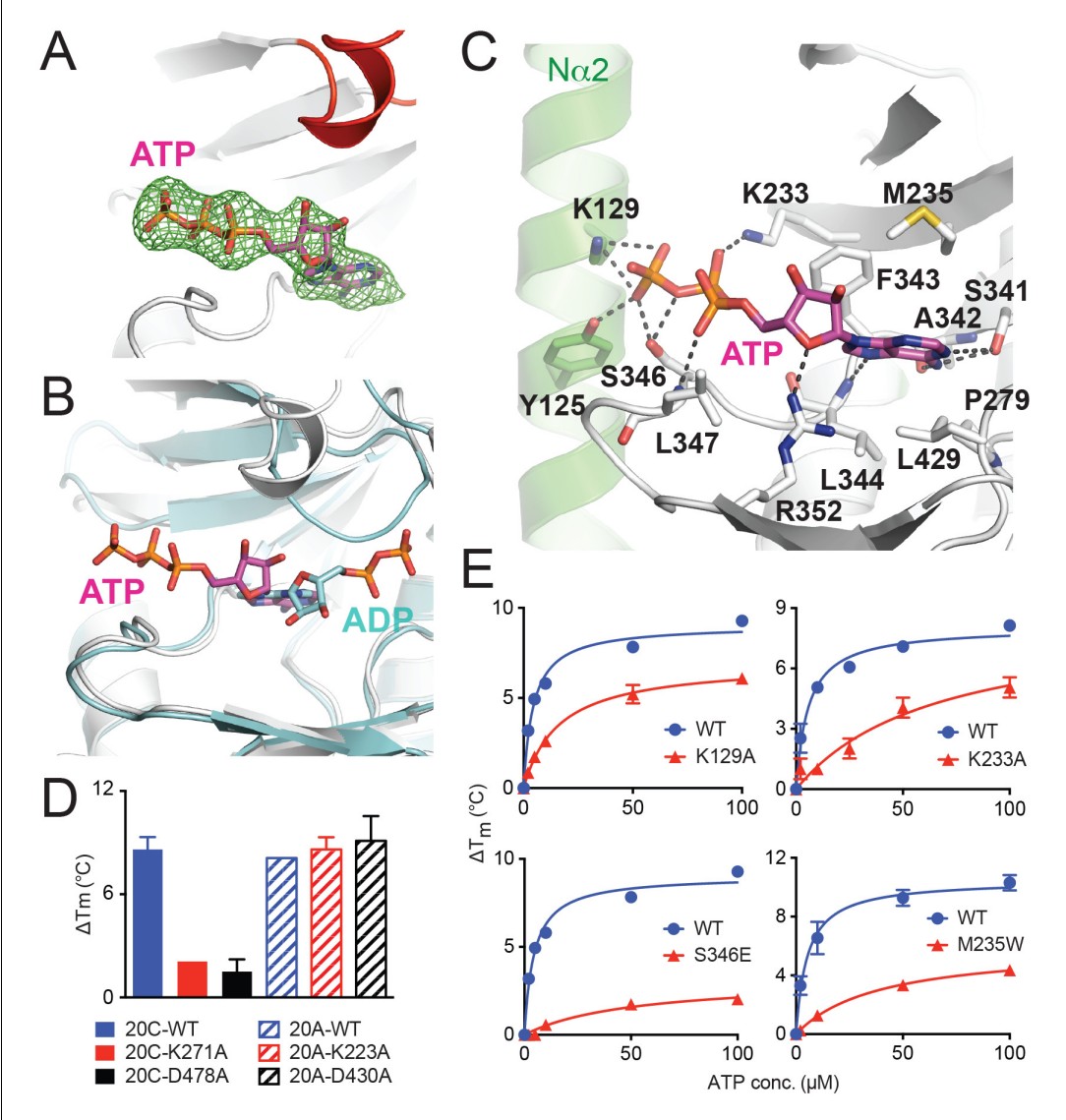

**Figure 3.** ATP binds to Fam20A in an inverted orientation. (**A**) The Fo-Fc difference electron density map (calculated before ATP is built and contoured at 3σ) is shown as a green mesh, revealing the presence of ATP. Fam20A is shown as ribbons and colored in the same scheme as *Figure 1A* (the Kβ1-Kβ2 loop in red). The ATP molecule built afterwards is shown in sticks, and its carbon atoms are colored in magenta. (**B**) Superimposition of Fam20A/ATP and ceFam20/ADP. Fam20A and ceFam20 are shown in white and cyan, respectively. The carbons in ATP and ADP are colored in magenta and cyan, respectively. (**C**) An enlarged image of the ATP-binding pocket of Fam20A showing the detailed molecular interactions with ATP. Fam20A is colored in the same scheme as in *Figure 1A*, with the Nα2 helix highlighted in green. Salt-bridge and hydrogen-bond interactions are shown as dashed lines. (**D**) Thermal stability shift assays of ATP binding to Fam20C or Fam20A. 100 µM and 250 µM ATP were used to assay Fam20A and Fam20C, respectively. 5 mM $MnCl_2$ was present in the Fam20C assay conditions. (**E**) Thermal stability shift assays of ATP binding to Fam20A WT, K129A, K233A, S346E, and M235W. Data were plotted and fitted as in *Figure 2B*.

In the Fam20A/ATP structure, the adenine moiety of ATP occupies a similar pocket as that of ADP in the ceFam20/ADP structure; however, the entire nucleotide is inverted in such a way that the ribose is turned 'upside down', and the phosphate groups point to the opposite direction, reaching to the Nα2 helix (*Figure 3B and C*). The adenine ring of ATP is sandwiched by several hydrophobic residues in Fam20A, including Met235 and Phe343 from the top, and Pro279, Leu344, and Leu429 from the bottom (*Figure 3C*). It also forms three hydrogen bonds with the side chain of Ser341 and the main chain atoms of Ala342 and Leu344. The ribose is coordinated by Arg352. The triphosphate

group, especially the γ-phosphate, is surrounded by extensive salt-bridge and hydrogen-bond interactions contributed by Tyr125 and Lys129 from the Nα2 helix, as well as Lys233, Ser346, and the main chain amide group of Leu347.

To confirm the inverted ATP-binding mode of Fam20A, we performed a series of mutagenesis experiments. We have previously shown that the adenine nucleotide binds to Fam20C in the conventional conformation, and is coordinated by a number of conserved residues, including Lys271[Fam20C] and Asp478[Fam20C] (*Xiao et al., 2013*). Lys271[Fam20C] is one of the two critical Lys that are involved in binding to the phosphate groups of the adenine nucleotide. Asp478[Fam20C] aligns with the DFG Asp in canonical kinases and is involved in binding to the metal ions that engage the phosphate groups of ATP. Indeed, neither Fam20C-K271A nor Fam20C-D478A mutant could bind ATP, as shown by the thermal stability shift assay (*Figure 3D*). This is consistent with our previous result showing that these mutants have greatly diminished kinase activity. In contrast, substitution of the corresponding residues in Fam20A (Lys223[Fam20A] and Asp430[Fam20A], respectively) with Ala has little effect on the ability of Fam20A to bind ATP (*Figure 3D*). Furthermore, we generated several Fam20A mutants to test the importance of residues involved in ATP-binding, including Fam20A-K129A, Fam20A-K233A, Fam20A-M235W, and Fam20A-S346E. Lys129[Fam20A] and Lys233[Fam20A] interact with the phosphate groups of ATP and help maintain the ATP in the inverted orientation (*Figure 3C*). In addition, substituting Met235[Fam20A] that is positioned near the ATP binding site with a bulkier Trp would interfere with adenine binding. Finally, mutating Ser346[Fam20A] to a negatively charged Glu would likely repel the phosphate groups of ATP. All four mutants showed attenuated ATP binding as compared to the wildtype protein (*Figure 3E*). Taken together, these mutation results corroborate our structural analyses, demonstrating that Fam20A binds ATP in a novel, inverted conformation that is fundamentally different from Fam20C and other kinases.

## Dimerization of Fam20A

Fam20A dimerizes/oligomerizes in solution in a concentration-dependent manner (*Figure 4A*). Also, V5-tagged Fam20A, but not Fam20B, co-immunoprecipitated with Flag-tagged Fam20A when co-expressed in U2OS cells, suggesting that Fam20A can form dimers/oligomers in the cell (*Figure 4B*). In the crystal, Fam20A forms a symmetrical, face-to-face dimer (*Figure 4C*). Each molecule in the dimer buries a ~2000 Å$^2$ interface, which accounts for 9.5% of its total solvent-accessible surfaces. The dimer is mediated by a number of hydrogen bonds and hydrophobic interactions between the two protomers (*Figure 4—figure supplement 1*). The Kβ1-Kβ2 loop from the two molecules facing each other, and the two pairs of short helices Kα2A and Kα2B are involved in forming a small four-helix bundle. The Kβ3-Kα3 loop of each protomer docks onto the C-lobe of the other protomer in the dimer (*Figure 4C*). In particular, Val249, Phe251, and Phe254 protrude out of the Kβ3-Kα3 loop to make strong hydrophobic contacts with the other molecule.

To determine whether the dimer interface seen in the crystal is important for Fam20A function, we substituted Val249, Phe251 and Phe254 with Ala (Fam20A-AAA). As demonstrated by the coimmunoprecipitation experiment, interaction between Flag-tagged Fam20A-AAA and V5-tagged Fam20A-AAA was significantly reduced compared to that between wild-type Fam20A (*Figure 4B*). Secretion of Fam20A-AAA was not obviously changed, suggesting that the mutations did not dramatically affect protein folding or stability (*Figure 4D*). Importantly, Fam20A-AAA showed impaired interaction with Fam20C (*Figure 4E*). Consistently, it failed to enhance Fam20C kinase activity in vitro (*Figure 4F and G*), nor could it increase Fam20C-catalyzed OPN phosphorylation in the cell (*Figure 4H*). Taken together, our results demonstrate that an intact dimer interface reported here is critical for Fam20A to bind Fam20C and allosterically enhance Fam20C activity.

## Discussion

The lumen of the endoplasmic reticulum as well as the Golgi apparatus is oxidative in nature. In order to adapt to this environment, secretory pathway proteins often form disulfide bonds to stabilize their structures. The crystal structure of Fam20A reveals two unique disulfide bonds as compared to Fam20C, which are mediated by a Fam20A-specific, evolutionarily conserved insertion. Remarkably, loss of this insertion due to abnormal mRNA splicing caused AI in a patient. Thus, compared to its close paralog Fam20C, Fam20A appears to have 'redesigned' its disulfide pattern to achieve a kinase-independent function.

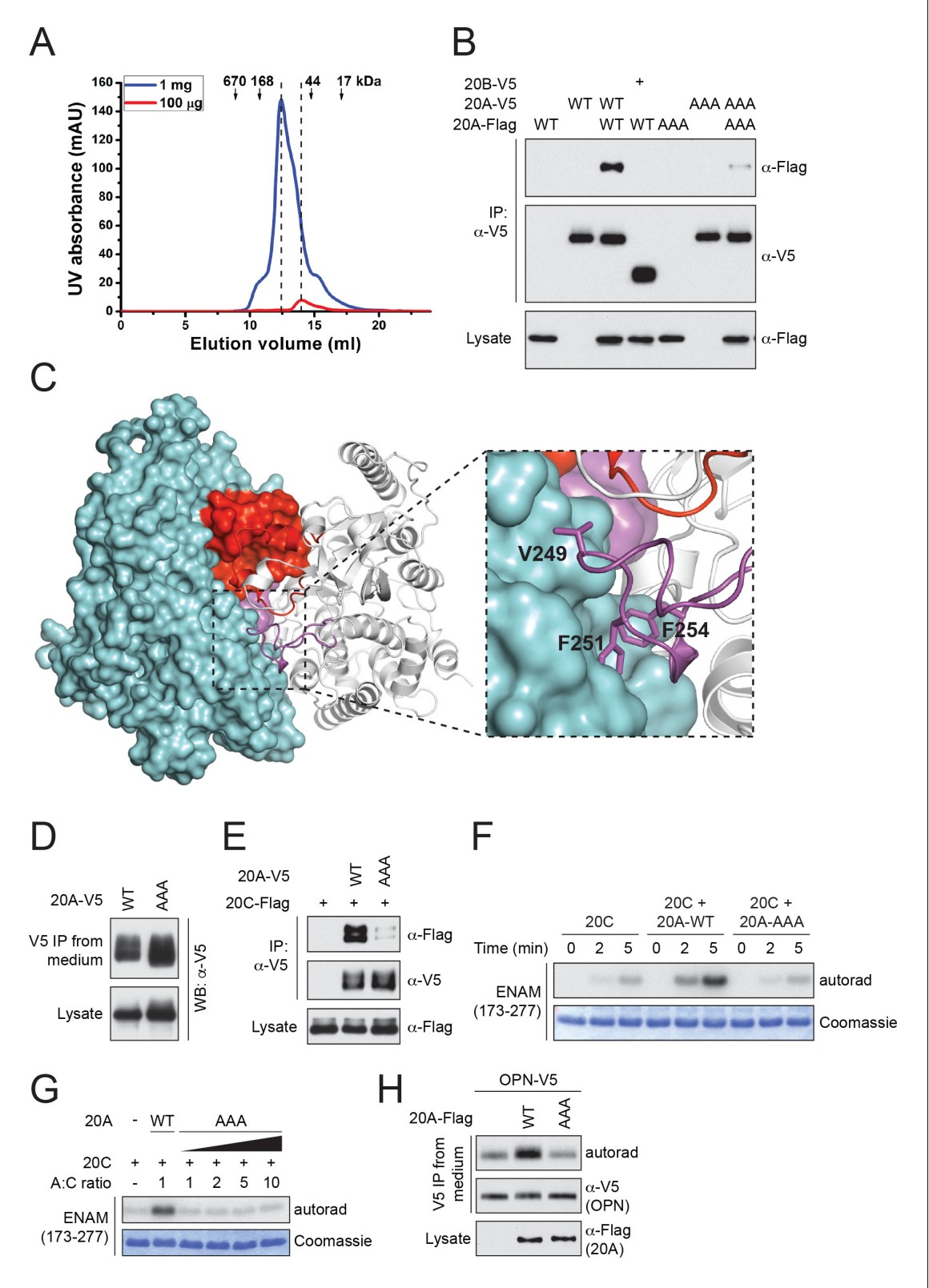

**Figure 4.** Fam20A forms a dimer in the crystal and an intact dimer interface is essential for its function. (**A**) Fam20A dimerizes in solution in a concentration-dependent manner. 1 mg or 100 μg purified Fam20A (both in 500 μl) were individually loaded on a Superdex 200 increase 10/300 GL column (GE healthcare) and eluted using 20 mM HEPES, pH 7.5, and 100 mM NaCl. The peak positions of the two runs are highlighted with dashed lines. The elution volumes of the molecular weight standards are indicated. (**B**) Co-immunoprecipitation of Fam20A-Flag and Fam20A-V5 (or Fam20B-

*Figure 4 continued on next page*

*Figure 4 continued*

V5) from U2OS cells. AAA, Fam20A V249A/F251A/F254A mutant. (**C**) The structure of Fam20A dimer. One protomer in the dimer is shown in ribbons, and the other is shown in surface representation. The Kβ1-Kβ2 and Kβ3-Kα3 loops are highlighted in red and magenta, respectively. The rest of the two molecules are shown in white and cyan. A close view of the Kβ3-Kα3 loop interaction is shown on the right. The side chains of Val249, Phe251, Phe254 are shown as sticks. (**D**) Secretion of Fam20A WT and V249A/F251A/F254A. (**E**) Effect of V249A/F251A/F254A mutation on Fam20A-Fam20C interaction. Fam20A-V5 (WT or AAA) and Fam20C-Flag were co-expressed in U2OS cells and coimmunoprecipitated from the cell lysate. (**F and G**) Effect of Fam20A-AAA on ENAM phosphorylation by Fam20C in vitro. (**H**) Effect of Fam20A-AAA on OPN phosphorylation in metabolically $^{32}$P radiolabeled U2OS cells.

The following figure supplement is available for figure 4:

**Figure supplement 1.** Schematic drawing of the interactions between the two protomers in the Fam20A dimer.

It is not uncommon for a pseudokinase to bind ATP. A number of pseudokinase structures have been determined with nucleotide bound, including HER3, STRADα, ILK, RNase L, and the JH2 domains of the JAK kinases (*Bandaranayake et al., 2012*; *Fukuda et al., 2009*; *Han et al., 2014*; *Huang et al., 2014*; *Jura et al., 2009*; *Min et al., 2015*; *Shi et al., 2010*; *Zeqiraj et al., 2009*; *Littlefield et al., 2014*). Among them, STRADα also binds ATP in a cation-independent manner. Notably, like Fam20A, STRADα also utilizes positively charged residues to engage the phosphate groups of ATP (*Zeqiraj et al., 2009*). This strategy could be used by other pseudokinases that bind nucleotides without cations, such as MLKL (*Murphy et al., 2014b*, *2013*, *2014a*). The Ca$^{2+}$/calmodulin-activated Ser-Thr kinase (CASK) also binds ATP without cations, although it appears to possess some catalytic activity and function as an atypical kinase rather than a pseudokinase (*Mukherjee et al., 2008*). The concentration of free cytoplasmic Mg$^{2+}$ is on the order of 1 mM, and it is generally believed that ATP exists in the cell mostly in a complex with Mg$^{2+}$. However, these studies suggest that free ATP also plays a role in various biological systems. A recent study has measured the cellular ATP and Mg$^{2+}$/ATP concentrations using $^{31}$P-NMR spectroscopy, and demonstrated that these two forms of ATP are present at ~54 μM and ~400 μM in the cytosol, respectively, and have a constant ratio of 0.135 (*Gout et al., 2014*). Although the concentrations of Mg$^{2+}$ (or Mn$^{2+}$), ATP, and Mg$^{2+}$/ATP (Mn$^{2+}$/ATP) in the secretory pathway are less clear, our results show that Fam20A can bind free ATP with a high affinity ($K_d^{app} \approx 3$ μM). Therefore, the divalent cation-independent ATP-binding of Fam20A is likely a physiologically relevant phenomenon.

What is unique about Fam20A is that it binds ATP in an inverted manner, which is first seen for a kinase. With the phosphates pointing away from the kinase domain, the Fam20A/ATP complex is unlikely catalytically competent. Nevertheless, our results show that ATP binding stabilizes Fam20A structure and improves the conformation homogeneity of the Fam20A/Fam20C complex. This is again reminiscent of the pseudokinase STRADα, which requires the binding of ATP to promote the assembly of the LKB1-MO25-STRADα complex and activate the kinase activity of LKB1 (*Zeqiraj et al., 2009*). Although binding of ATP to Fam20A results in few conformational changes in the crystal structures we reported here, the presence of ATP greatly increases the melting temperature of Fam20A. It is likely that without ATP, the structure of Fam20A is more flexible and less stable. Binding of ATP may help stabilize the conformation of Fam20A by mediating additional interactions between the kinase core and the N-terminal segment.

The Fam20A/Fam20C complex appears to be a tetramer that contains two molecules of Fam20A and two molecules of Fam20C (*Cui et al., 2015*). Here we show that Fam20A forms a dimer in the crystal, and residues in the dimer interface are critical for Fam20A to bind and activate Fam20C. At this stage, it remains unclear whether this Fam20A dimer also exists in the Fam20A/Fam20C tetramer. It is possible that the Fam20A dimer seen here serves as a platform to allow the docking of Fam20C and assembly of the complex. However, we cannot rule out the possibility that the Fam20A dimer in the crystal is not physiological, and the surface mediating Fam20A dimerization is actually used to interact with Fam20C directly. A high-resolution Fam20A/Fam20C complex structure is ultimately needed to elucidate how Fam20A interacts with Fam20C and allosterically regulates Fam20C activity.

In conclusion, we have solved the crystal structure of Fam20A, a secretory pathway pseudokinase. We have identified a novel mode of ATP binding and a unique disulfide pattern that is indispensable

for Fam20A function. Our results expand the diversity of protein kinases and shed light on the molecular pathology of diseases caused by mutations in Fam20A.

## Materials and methods

### Protein expression and purification

DNA fragment encoding Human Fam20A residues 69–529 was cloned into a modified pI-secSUMOstar vector (LifeSensors, Malvern, PA) containing a tobacco etch virus (TEV) protease cleavage site. Bacmids were generated using the Bac-to-Bac system (Invitrogen, Carlsbad, CA). Recombinant baculoviruses were generated and amplified using the Sf21 insect cells (RRID: CVCL_0518), maintained in the SIM SF medium (Sino Biological Inc., Beijing, China). For protein production, Hi5 cells (RRID: CVCL_C190) grown in the SIM HF medium (Sino Biological Inc.) were infected at a density of 1.5–2.0 $\times$ $10^6$ cells/ml. 48 hr post infection, 2 liters of conditioned medium were collected by centrifugation at 200x g. The medium was concentrated using a Hydrosart Ultrafilter (Sartorius, Göttingen, Germany) and exchanged into the binding buffer containing 25 mM Tris-HCl, pH 8.0, 200 mM NaCl. The proteins were then purified using the Ni-NTA resin (GE healthcare, Chicago, IL). The N-terminal 6xHis-SUMO fusion tag was removed by the TEV protease. Untagged Fam20A was further purified by the anion exchange chromatography using a Resource Q column (GE healthcare), followed by the size-exclusion chromatography using a Superdex 200 increase 10/300 GL column (GE healthcare).

### Thermal stability shift assay

For the thermal stability shift assays, proteins were diluted to a final concentration of 2 µM in a buffer containing 20 mM HEPES, pH 7.5, 100 mM NaCl. Different concentrations of nucleotides or metal ions were added as indicated. SYPRO Orange dye (Molecular Probes, Eugene, OR; S6650) was added to a final concentration of 5$\times$ to monitor protein denaturation. Thermal scanning (25–85°C at 0.6 °C/min) was performed using a LightCycler 480 System (Roche, Pleasanton, CA) with the combination filters of 465 nm (excitation) and 580 nm (emission). Data were analyzed using the software provided by the manufacture.

### Crystallization

Fam20A in 20 mM HEPES, pH 7.5, 100 mM NaCl was concentrated to 7 mg/ml and used for crystallization. The crystals were grown at 20°C using the hanging-drop vapor-diffusion method. The precipitant solution contained 1.7 M ammonium sulfate, 0.1 M HEPES, pH 7.5, and 6% (w/v) PEG 400. The crystals reached full size in 10–14 days, and were then transferred into a cryo-protection solution containing 20% (v/v) glycerol, 1.7 M ammonium sulfate, 0.1 M HEPES, pH 7.5, and 6% (w/v) PEG 400 and flash-frozen in liquid nitrogen. To obtain the ATP-bound structure, both co-crystallization and soaking experiments were performed. For co-crystallization, Fam20A protein solution was supplemented with 20 mM ATP before mixed with above precipitant solution and crystallized. For soaking, the apo crystal obtained above was soaked in the cryo-protection solution supplemented with 10 mM ATP for 1 hr before frozen.

### Data collection and structure determination

Diffraction data were collected at Shanghai Synchrotron Radiation Facility beamline BL17U and processed with HKL2000 (HKL Research). The structure of Fam20A was determined by molecular replacement using the ceFam20 structure as the search model in Phaser (*McCoy et al., 2007*), RRID: SCR_014219). The structural model was then manually built in Coot (*Emsley et al., 2010*), RRID: SCR_014222) and refined with Phenix (*Adams et al., 2010*), RRID: SCR_014224). Five percent randomly selected reflections were used for cross-validation (*Brünger et al., 1998*).

### Bioinformatics and structural analysis

Multiple sequence alignment was performed using PROMALS3D (*Pei et al., 2008*). Structural alignment between Fam20A and ceFam20 was performed using Dali server (*Holm and Rosenström, 2010*), RRID: SCR_013433). Interaction between the two protomers in the Fam20A dimer was

analyzed using DIMPLOT (*Laskowski and Swindells, 2011*). Molecular graphics were prepared using PyMol (Schrödinger, LLC., RRID: SCR_000305).

## Negative stain grid preparation and electron microscopy

FAM20A/20C complexes were mixed with or without ATP and further purified using size exclusion chromatography. After determining the concentration of peak fractions, we diluted the sample in gel filtration buffer to 40 nM and incubated on continuous carbon grids (Electron Microscopy Sciences, Hatfield, PA; CF400-Cu) for 30 s. After incubation, the grid was transferred directly onto 5 × 75 μl droplets of 2% uranyl acetate and then blotted dry.

Negative stain grids were imaged on an Tecnai Sphera (FEI Company, Hillsboro, OR) at 200 kV using a US4000 CCD detector (Gatan Inc., Pleaston, CA) with the Leginon automated data collection software (*Suloway et al., 2005*) over a defocus range of 1–2.5 μm and a dose of 40 e-/$\text{Å}^2$ at a nominal magnification of 62,000X (1.90 Å/pix). 93 micrographs were collected from the –ATP grid and 88 micrographs were collected from the +ATP grid, representative micrographs are shown in *Figure 2—figure supplement 2*.

## Single particle image analysis

The Appion processing pipeline was used for particle picking and extraction, as well as CTF estimation (*Lander et al., 2009*). For each dataset, particles were initially picked using DoG Picker (*Voss et al., 2009*), extracted at a pixel size of 3.80 Å/pix (64 × 64 pixels), and subjected to 2D classification using Imagic (*van Heel et al., 1996*). From the resulting class averages, representative class averages were used to re-pick particles using template-based picking with FindEM (*Roseman, 2004*). The CTF for each micrograph was estimated using CTFFIND4 (*Rohou and Grigorieff, 2015*). Particles were extracted from phase-flipped micrographs and binned to a pixel size of 3.80 Å/pix (64 × 64 pixels). This yielded 6051 particles for the 'ATP-' dataset and 9116 particles for the 'ATP+' dataset. 2D class averages shown in *Figure 2—figure supplement 1* were calculated from iterative rounds of 2D classification and alignment using Imagic (*van Heel et al., 1996*) to finish with approximately 100 particles per 2D average.

For the alignment of 'ATP−' and 'ATP+' class averages in *Figure 2F*, SPIDER (*Frank et al., 1996*) was used to find the highest cross correlation score between class averages from each dataset. Following this comparison, the best matching pair were aligned and shown in *Figure 2F* to illustrate the overall size difference between the two datasets.

## Others

In vitro kinase assay, mammalian cell culture, transfection, immunoprecipitation, 32P orthophosphate metabolic labeling, as well as the antibodies used in this study were described previously (*Cui et al., 2015*). U2OS cells (RRID: CVCL_0042) were originally obtained from and authenticated by the American Type Culture Collection (ATCC, Manassas, VA). Rabbit anti-V5 polyclonal antibody (Millipore, Billerica, MA; AB3792; RRID: AB_91591) was used for immunoprecipitation. Monoclonal anti-V5 antibody (Life Technologies, Carlsbad, CA; R960-25; RRID: AB_2556564) and anti-FLAG antibody (Sigma, St. Louis, MO; F3165; RRID: AB_259529) were used for Western blotting.

## Acknowledgements

We are grateful to staff members of Shanghai Synchrotron Radiation Facility (beamline BL17U) for assistance in X-ray data collection. We thank Brenden Park and Natalie Chen for their technical support, and Hongwei Guo for the access of the Real-Time PCR machine. We thank Carolyn Worby, Xing Guo, Ping Zhang, and members of the Xiao and Dixon laboratories for insightful discussions and comments regarding the manuscript.

## Additional information

### Funding

| Funder | Grant reference number | Author |
|---|---|---|
| National Natural Science Foundation of China | 31570735 | Junyu Xiao |
| National Institutes of Health | DK018849 | Jack E Dixon |
| National Institutes of Health | DK018024 | Jack E Dixon |
| Human Frontier Science Program | LT000659/2013-L | Jixin Cui |
| Damon Runyon Cancer Research Foundation | DRG 2171-13 | Michael A Cianfrocco |
| Howard Hughes Medical Institute | | Jack E Dixon |
| National Key Research and Development Plan | 2016YFC0906000 | Junyu Xiao |

The funders had no role in study design, data collection and interpretation, or the decision to submit the work for publication.

### Author contributions

JC, Data curation, Writing—original draft, Writing—review and editing; QZ, HZ, Data curation, Writing—review and editing; MAC, Data curation, Writing—original draft; AEL, Supervision, Funding acquisition; JED, Supervision, Funding acquisition, Writing—review and editing; JX, Data curation, Funding acquisition, Writing—original draft, Writing—review and editing

### Author ORCIDs

Junyu Xiao, http://orcid.org/0000-0003-1822-1701

## Additional files

### Major datasets

The following datasets were generated:

| Author(s) | Year | Dataset title | Dataset URL | Database, license, and accessibility information |
|---|---|---|---|---|
| Cui J, Zhu Q, Zhang H, Cianfrocco MA, Leschziner AE, Dixon JE, Xiao J | 2016 | Crystal structure of Fam20A | http://www.rcsb.org/pdb/explore/explore.do?structureId=5WRR | Publicly available at the RCSB Protein Data Bank (accession no: 5WRR) |
| Cui J, Zhu Q, Zhang H, Cianfrocco MA, Leschziner AE, Dixon JE, Xiao J | 2016 | Crystal Structure of Fam20A in complex with ATP | http://www.rcsb.org/pdb/explore/explore.do?structureId=5WRS | Publicly available at the RCSB Protein Data Bank (accession no: 5WRS) |

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
