## [Decision Letter]

Thank you for submitting your article "Structure of Fam20A reveals a pseudokinase featuring unique disulfide pattern and inverted ATP-binding" for consideration by *eLife*. Your article has been reviewed by three peer reviewers, and the evaluation has been overseen by a Reviewing Editor and Tony Hunter as the Senior Editor. The following individual involved in review of your submission has agreed to reveal his identity: Dario R Alessi (Reviewer #1).

The reviewers have discussed the reviews with one another and the Reviewing Editor has drafted this decision to help you prepare a revised submission.

Summary:

In this paper, the authors have extended their work on the Fam20A pseudokinase, a secreted protein they had previously shown to bind to and activate the secreted Fam20C kinase. Here, they report the crystal structure of Fam20A bound to ATP, which forms a dimer and reveals how a 17 aa insert and a set of S-S bonds distinct from those in Fam20C distort the Fam20A active site precluding phosphotransferase activity, and also result in ATP binding in the absence of metal ions in an orientation distinct from the way in which ATP binds productively to the active site of Fam20C. They went on to show that ATP stabilizes Fam20A and also alters the conformation of the Fam20A/Fam20C heterotetramer. They also defined and validated the Fam20A dimer interface, and demonstrated that this is important for activating Fam20C.

This is a nice follow up to their previous 2015 *eLife* paper, where they reported that Fam20A stimulates the activity of Fam20C, and their new results provide important insights into how pseudokinases can play a functional role in a phosphorylation pathway, and also come to the surprising conclusion that ATP binds Fam20A in an orientation different from the way ATP binds to most active protein kinases.

Essential revisions:

1) The reviewers have general concerns about the deduced binding orientation of ATP in the nucleotide binding pocket, which is an orientation that has never been observed previously for any protein kinase or pseudokinase. Although some mutagenesis was performed to corroborate the novel binding mode (and is generally supportive), one issue here is that ATP was soaked into the crystals to obtain the ATP-bound structure, raising the possibility that crystal packing prevented ATP binding in the normal mode. Curiously, the β2 and β3 lysines, which interact with the phosphate groups of ATP in Fam20C (ADP in crystal structure), are conserved in Fam20A (Lys223 and Lys237), even though they would have no direct binding role in this non-conventional ATP binding mode. Because much of the significance of this study hinges on the novel ATP binding mode, it is important that the authors either attempt to co-crystallize ATP with Fam20A, or, if they already tried this and failed, mention this in the manuscript and state that the Fam20A dimer interface (or any other lattice interaction) would not be expected to preclude binding of ATP in the conventional mode. Additional (easily obtainable) mutagenesis data, particularly for Lys223 and Lys237, to bolster their claim of a novel ATP binding mode for Fam20A would strengthen their conclusions. Although the two mutations they did test, M235W and S346E, impaired ATP binding and the interaction with Fam20C, this could have been the result of a loss of stability unrelated to ATP binding. Another possible way of defining the ATP binding orientation would be to determine whether FSBA treatment leads to labeling of Fam20A in solution (Lys 237 in the "VAXK" motif would be expected to react, if FSBA bound in the conventional ATP binding mode).

2) The authors claim that Fam20A may bind nucleotides independently of divalent cations in vivo. The presence of millimolar levels of magnesium within cells would imply that almost all of the cellular ATP will be complexed to magnesium, raising the issue of whether there is any ATP in the cell free of divalent metal ions available to bind to Fam20A. The authors should comment on this. In this regard, is there any way for the authors to demonstrate whether the ability of Fam20A to activate Fam20C in vitro is affected by whether Fam20A is complexed to Mg ATP or metal-free ATP? We appreciate that this experiment might be technically challenging as Mg-ATP is needed to assay Fam20C.

3) The authors demonstrate that Fam20A binds GTP and other nucleotides including ADP. Here again it would be interesting, if feasible, if the authors could ascertain how these different Fam20A nucleotide complexes, including ADP, impact on the ability of Fam20A to activate Fam20C in protein kinase activity assays.

4) The data in Figure 3 showing that the S346E mutant that has 10-fold reduced affinity for ATP but still activates Fam20C is potentially confusing. The authors speculate that the S346E mutant may induce a conformation that mimics the ATP-induced conformation of Fam20A to account for this result. However, no data are presented to support this conclusion. It would be better if the authors could design at least one additional non-ATP binding mutant that does not adopt the nucleotide bound conformation to really demonstrate that abolishing ATP binding has a major impact on Fam20C activation. In addition, they could also mutate the additional residues in the S346E Fam20C mutant that they argue compensate for lack of ATP binding of this mutant. Can the authors also confirm that ATP concentrations used in the Fam20C kinase assay are sufficiently low that the S346E mutant would not be able to bind ATP?

5) The authors should discuss whether other kinases or pseudokinases have been identified that bind ATP in the equivalent inverted orientation as Fam20A? Can this inverted binding be linked to any evolutionary hypothesis or ancestor protein that these ATP binding proteins might have evolved?

---

## [Author Response]

*Essential revisions:*

*1) The reviewers have general concerns about the deduced binding orientation of ATP in the nucleotide binding pocket, which is an orientation that has never been observed previously for any protein kinase or pseudokinase. Although some mutagenesis was performed to corroborate the novel binding mode (and is generally supportive), one issue here is that ATP was soaked into the crystals to obtain the ATP-bound structure, raising the possibility that crystal packing prevented ATP binding in the normal mode. Curiously, the β2 and β3 lysines, which interact with the phosphate groups of ATP in Fam20C (ADP in crystal structure), are conserved in Fam20A (Lys223 and Lys237), even though they would have no direct binding role in this non-conventional ATP binding mode. Because much of the significance of this study hinges on the novel ATP binding mode, it is important that the authors either attempt to co-crystallize ATP with Fam20A, or, if they already tried this and failed, mention this in the manuscript and state that the Fam20A dimer interface (or any other lattice interaction) would not be expected to preclude binding of ATP in the conventional mode. Additional (easily obtainable) mutagenesis data, particularly for Lys223 and Lys237, to bolster their claim of a novel ATP binding mode for Fam20A would strengthen their conclusions. Although the two mutations they did test, M235W and S346E, impaired ATP binding and the interaction with Fam20C, this could have been the result of a loss of stability unrelated to ATP binding. Another possible way of defining the ATP binding orientation would be to determine whether FSBA treatment leads to labeling of Fam20A in solution (Lys 237 in the "VAXK" motif would be expected to react, if FSBA bound in the conventional ATP binding mode).*

A) Although not mentioned in the original manuscript, we had co-crystallized ATP with Fam20A and determined the structure at 2.9 Å. In the co-crystal structure, ATP is present in the same orientation as that in the soaking experiment and is therefore not an artifact caused by the soaking experiment. The crystal obtained by soaking diffracted slightly better (2.75 Å), and that is why it was presented in the manuscript. We have now described these experimental details in the revised manuscript (subsection “ATP binds to Fam20A in an inverted orientation”, first paragraph, and subsection “Crystallization”).

B) To further confirm the inverted ATP-binding mode of Fam20A, we performed additional mutagenesis experiments as suggested. We have previously shown that the adenine nucleotide binds to Fam20C in the conventional conformation, and is coordinated by a number of conserved residues (Xiao et al., PNAS, 2013). For example, Fam20C-K271 is one of the two critical Lys that are involved in binding to the phosphate groups of the adenine nucleotide. Indeed, the Fam20C-K271A mutant could not bind ATP, as shown by the thermal stability shift assay (Figure 3). This is consistent with our previous result to show that this mutant has greatly diminished kinase activity. In contrast, mutating the corresponding Lys in Fam20A (K223) to Ala has no effect on ATP binding (Figure 3). Fam20C-D478 corresponds to the DFG Asp in conventional kinases and is involved in binding to the metal ions that engage the phosphate groups of ATP. The Fam20C-D478A mutant also failed to bind ATP (Figure 3). However, substitution of the corresponding Asp in Fam20A (D430) to Ala has little effect on the ability of Fam20A to bind ATP (Figure 3). Furthermore, we also tested the function of ATP-binding residues that are uniquely present in Fam20A by generating two single mutants: Fam20A-K129A and Fam20A-K233A. K129 and K233 interact with the phosphate groups of ATP and help maintain the ATP in a flipped orientation (Figure 3), and both mutants showed attenuated ATP binding as compared to the wildtype protein (Figure 3 in the revised manuscript). These additional mutation experiments, together with the data we presented previously for M235W and S346E, corroborate our structural analyses, demonstrating that Fam20A binds ATP in a novel, inverted conformation that is fundamentally different from Fam20C and other kinases.

C) Regarding the stability of the Fam20A-M235W and Fam20A-S346E, we think the structural integrity of these proteins was not greatly affected for two reasons: First, protein folding is under stringent surveillance within the secretory pathway. Usually, only properly folded protein can be secreted effectively. The secretion of Fam20A-M235W and Fam20A-S346E were similar to wildtype Fam20A, indicating that these mutants are likely well folded. Second, the melting temperatures of these mutants are very close to that of WT Fam20A in our thermal stability shift assay, suggesting that these mutants are as stable as the wildtype protein. Nevertheless, in the revised manuscript, we did not include data related to Fam20C activation by these two mutants due to reasons elaborated in our response to the point #4 below.

D) The FSBA labeling experiment is likely a tricky experiment, since FSBA is very reactive and can label residues that are far away from the ATP-binding site. For example, Paynet et al. tried to map the ATP-binding residues in human fructosamine 3-kinase-related protein using this technique (Biochem J., 2008. doi: 10.1042/BJ20080389). Among the five residues that are labeled by FSBA in their study, one (Lys269) is located very far away from the ATP-binding pocket, and two (Tyr33, His90) are located on the opposite side of where the phosphate groups are expected to bind. It therefore appears to us that it will be very difficult to define the ATP-binding orientation in Fam20A using this technique, so we did not perform the FSBA labeling experiment. We hope this can be understood by the reviewer. On the other hand, the additional mutation data we presented above are all consistent with the novel ATP orientation revealed by the crystal structure.

*2) The authors claim that Fam20A may bind nucleotides independently of divalent cations* in vivo*. The presence of millimolar levels of magnesium within cells would imply that almost all of the cellular ATP will be complexed to magnesium, raising the issue of whether there is any ATP in the cell free of divalent metal ions available to bind to Fam20A. The authors should comment on this. In this regard, is there any way for the authors to demonstrate whether the ability of Fam20A to activate Fam20C* in vitro *is affected by whether Fam20A is complexed to Mg ATP or metal-free ATP? We appreciate that this experiment might be technically challenging as Mg-ATP is needed to assay Fam20C.*

A)The concentration of free cytoplasmic Mg^2+^ is on the order of 1 mM, and it is generally believed that ATP exists in the cell mostly in a complex with Mg^2+^. However, multiple studies suggest that free ATP also plays a role in various biological systems. For example, STRADα, a cytoplasmic pseudokinase and allosteric activator of LKB1, has been shown to bind ATP without cations. A recent study has measured the cellular ATP and Mg^2+^/ATP concentrations using ^31^P-NMR spectroscopy, and demonstrated that these two forms of ATP are present at ~54 µM and ~400 µM in the cytosol, respectively, and have a constant ratio of 0.135 (Gout et al., *PNAS*, 2014. DOI: 10.1073/pnas.1406251111). Although the concentrations of Mg^2+^ (or Mn^2+^), ATP, and Mg^2+^/ATP (Mn^2+^/ATP) in the lumen of the ER and Golgi apparatus are less clear, our results show that Fam20A can bind free ATP with a high affinity (K_d_^app^ ≈ 3 µM). Therefore, the divalent cation independent ATP-binding of Fam20A is likely a physiologically relevant phenomenon. We have included these discussions in the revised manuscript (Discussion, second paragraph).

B) As the reviewer pointed out, it is indeed technically challenging to test how the ability of Fam20A to activate Fam20C is affected by metal ions. Because Mn^2+^ is required for the kinase reaction of Fam20C (Tagliabracci et al., Science, 2012), it is difficult to unravel the interplay of Mn^2+^ and ATP in a system containing both Fam20A (that prefers to bind ATP without Mn^2+^) and Fam20C (that requires Mn^2+^/ATP for optimal activity). We thank the reviewers for appreciating this technical difficulty.

*3) The authors demonstrate that Fam20A binds GTP and other nucleotides including ADP. Here again it would be interesting, if feasible, if the authors could ascertain how these different Fam20A nucleotide complexes, including ADP, impact on the ability of Fam20A to activate Fam20C in protein kinase activity assays.*

For similar reasons stated above, it is technically challenging to perform these experiments. Fam20A can bind ATP, and to a lesser degree, GTP and ADP (Figure 2), it is therefore difficult to make sure that Fam20A is in a GTP/ADP-bound form in a kinase reaction containing both ATP and GTP/ADP, and Mn^2+^. Furthermore, excessive GTP and ADP may inhibit the activity of Fam20C, due to their competition with ATP to bind to metal ions and Fam20C. Therefore, it is challenging to test the effect of these compounds on the ability of Fam20A to activate Fam20C.

*4) The data in Figure 3 showing that the S346E mutant that has 10-fold reduced affinity for ATP but still activates Fam20C is potentially confusing. The authors speculate that the S346E mutant may induce a conformation that mimics the ATP-induced conformation of Fam20A to account for this result. However, no data are presented to support this conclusion. It would be better if the authors could design at least one additional non-ATP binding mutant that does not adopt the nucleotide bound conformation to really demonstrate that abolishing ATP binding has a major impact on Fam20C activation.*

As the reviewers suggested, we also attempted to generate additional non-ATP binding mutants by making K129A and K233A mutants of Fam20A. As mentioned in the answer to point #1, these two mutants showed a moderate decline of ATP binding, but both of them were still able to enhance Fam20C activity. Again, this is likely due to the fact that other positively charged residues can still bind the phosphate groups of ATP when K129/K233 are mutated.

To avoid over-interpreting our data, we have decided to leave out the kinase assay results present in the original Figure 3 and also remove the speculation on the S346E mutant from the manuscript. Nevertheless, this does not affect the major conclusions of this paper, including that ATP greatly stabilizes Fam20A structure and the Fam20A/Fam20C complex, as shown by the significant increase of Fam20A melting temperature in the presence of ATP and the gel filtration analyses on the Fam20A/Fam20C complex. It is increasingly clear that protein stability of Fam20A is critical for its function. A series of loss-of-function point mutations identified from patients with Enamel-Renal syndrome, as well as the Δexon3 internal deletion described in this manuscript, all destabilize Fam20A. Binding ATP appears to be a mechanism for Fam20A to reinforce its structural stability.

Fam20AATP bindingFam20C activationWTStrong+M235WLow-K129AModerate+K233AModerate+S346ELow+S346E/Y125A/K129ALow+

*In addition, they could also mutate the additional residues in the S346E Fam20C mutant that they argue compensate for lack of ATP binding of this mutant.*

As per the reviewers’ suggestion to mutate the compensating residues in Fam20A-S346E, we generated a triple mutant Fam20A-S346E/Y125A/K129A. We hypothesized that in Fam20A-S346E, the glutamate may contact Y125 and K129 (Figure 3) to mimic an ATP-bound conformation. However, this triple mutant can still activate Fam20C, although its ATP-binding ability is significantly diminished (see the table above). A close examination of Fam20A structure reveals that the region surrounding the γ-phosphate group of ATP is highly positively charged, with K233 (Figure 3), R227, and R132 all present in close vicinity. The glutamate may also interact with these residues to stabilize the Fam20A structure. Probably more residues need to be mutated in order to disrupt the interaction between E346 and the rest of the protein. However, we did not proceed on another round of mutagenesis study due to the time limit (It takes about a month to generate a mutant protein in our baculovirus/insect cell based system).

*Can the authors also confirm that ATP concentrations used in the Fam20C kinase assay are sufficiently low that the S346E mutant would not be able to bind ATP?*

The K_d_^app^ of Fam20A-S346E for ATP is ~50 μM, and we used 100 μM ATP previously in the kinase assay. In order to make sure ATP concentration is low enough so that Fam20A-S346E does not bind ATP, we have repeated the kinase assay using 10 μM ATP (and 10 μM of Mn^2+^). Fam20A-S346E can still efficiently activate Fam20C under this condition (Figure 5). This result suggests that the ability of Fam20A-S346E to activate Fam20C is likely not due to its residual ATP-binding capability.

Author response image 1.**DOI:**
http://dx.doi.org/10.7554/eLife.23990.012

*5) The authors should discuss whether other kinases or pseudokinases have been identified that bind ATP in the equivalent inverted orientation as Fam20A? Can this inverted binding be linked to any evolutionary hypothesis or ancestor protein that these ATP binding proteins might have evolved?*

To the best of our knowledge, no other kinases or pseudokinases bind ATP in the inverted orientation similarly to Fam20A. This has been pointed out in the third paragraph of the Discussion.

It appears that Fam20A is derived from Fam20C during evolution, because (1) Fam20A orthologs only exist in vertebrates, while Fam20C orthologs can be found in vertebrates as well as invertebrates, such as *C. elegans* and *Drosophila*, and (2) Fam20C is the closest paralog of Fam20A. The ability of Fam20A to bind ATP in an inverted orientation appears to be a new trait that Fam20A obtained during evolution.